# Alarming Antibiotic Resistance of Lactobacilli Isolated from Probiotic Preparations and Dietary Supplements

**DOI:** 10.3390/antibiotics11111557

**Published:** 2022-11-05

**Authors:** Elizaveta Anisimova, Islamiya Gorokhova, Guzel Karimullina, Dina Yarullina

**Affiliations:** Department of Microbiology, Kazan Federal University, Kremlevskaya Str. 18, 420008 Kazan, Russia

**Keywords:** Lactobacilli, probiotics, safety, antibiotic resistance, resistance genes

## Abstract

In this study, we screened eight commercially available brands of *Lactobacillus*-containing probiotic preparations and dietary supplements for resistance towards commonly administered antibiotics of different classes. According to disc diffusion results, most of the isolates were resistant to vancomycin and susceptible to penicillin-type antibiotics (ampicillin and amoxicillin), carbapenems (imipenem, meropenem, and ertapenem), and inhibitors of protein synthesis (chloramphenicol, erythromycin, tetracycline, clarithromycin, and linezolid). However, based on minimum inhibitory concentration (MIC) values, six strains were reconsidered as resistant to tetracycline. All tested lactobacilli were resistant towards amikacin, ciprofloxacin, and norfloxacin. Resistance to cephalosporins was highly variable and decreased in the following order: ceftazidime/cefepime, ceftriaxone, cefotaxime, cefazolin, and cefoperazone. PCR screening for antibiotic resistance determinants in probiotic lactobacilli revealed a wide occurrence of vancomycin resistance gene *van*X, ciprofloxacin resistance gene *par*C, and extended-spectrum β-lactamase gene *bla*TEM. We also detected the *tet*K gene for tetracycline resistance in one isolate. Additionally, we identified discrepancies between the claims of the manufacturers and the identified species composition, as well as the enumerated amount of viable bacteria, for several products. The results of this study raise concerns about the safety of lactobacilli for human consumption as probiotics, as they may act as reservoirs of transferable antibiotic resistance genes.

## 1. Introduction

The genus *Lactobacillus* has significant scientific and economic value. Although it was recently reclassified into 25 genera [1], for the purpose of this paper, “*Lactobacillus*” refers to the former genus. Lactobacilli are widely used in the food industry for food production and preservation. They are being explored as starters for dairy products, fermented vegetables, sausages, bread, beer, and other fermented beverages, as well as silage cultures [2]. *Lactobacilli* play an essential role as probiotics, which are defined as “live microorganisms, which when administered in adequate amounts, confer a health benefit on the host” [3]. Probiotic lactobacilli are marketed as pharmaceuticals and dietary supplements but can also be found in foods, such as yogurt and other dairy products [4]. The application of *Lactobacillus* as probiotics is mainly based on their positive role in the human normal intestinal microbiota. Lactobacilli exert their beneficial effects on the host’s health through the following major mechanisms: antagonistic activity toward pathogens, favorable alteration of the host microbiota composition, and modulation of immune responses [5]. Specific *Lactobacillus* strains have also been demonstrated to improve the metabolism of dietary components, inactivate toxic and mutagenic compounds, reduce serum cholesterol level, produce vitamins, exert antioxidant activity, and display many other abilities. Several experimental and clinical studies have confirmed their efficiency in prevention and/or treatment of gastrointestinal disorders, such as diarrhea, colitis, constipation, irritable bowel syndrome, and colorectal cancer. Moreover, *Lactobacillus* has shown promising effects against a number of pathologies, including asthma, eczema, obesity, atherosclerosis, cancer, autism spectrum disorder, and depression [6]. It is therefore no surprise that *Lactobacillus*-based probiotics are appealing to consumers, furthering the growth of the global probiotic market, which is projected to reach a value of USD 74.69 billion by the end of 2025 [7]. Despite of the tremendous economic exploitation of lactobacilli, concerns about their safety have not been adequately addressed. Endocarditis, bacteremia, and other infections caused by lactobacilli have been reported [8]. However, some *Lactobacillus* species have been conferred qualified presumption of safety (QPS) status by the European Food Safety Authority (EFSA) [9], and others are generally recognized as safe (GRAS) by the U.S. Food and Drug Administration (FDA) [10]. In recent years, lactobacilli have received a considerable amount of attention, owing to their potential involvement in the spread of antibiotic resistance. Intrinsic (as opposed to acquired) resistance has a minimal potential for horizontal spread and therefore usually poses no risk in non-pathogenic bacteria [11]. Most lactobacilli show intrinsic resistance to aminoglycosides (gentamicin, kanamycin, neomycin, and streptomycin), vancomycin, ciprofloxacin, and trimethoprim. *Lactobacillus* species are generally susceptible to β-lactams (penicillin and ampicillin) and inhibitors of protein synthesis (tetracycline, erythromycin, chloramphenicol, and linezolid) [11,12,13,14,15]. However, acquired resistance to tetracycline, erythromycin, chloramphenicol, and clindamycin has been detected in lactobacilli isolated from various sources [12,13,16]. Resistance transfer from lactobacilli to other organisms has been demonstrated both in vitro and in vivo [16,17]. However, given the current regulatory deficits of the probiotic industry [18], common probiotic species are used without consideration of their genetic instability and possible harboring of acquired or transmissible antibiotic resistance determinants.

The aim of this study was to characterize phenotypic and genotypic antibiotic resistance profiles of lactobacilli isolated from commercial probiotics.

## 2. Results

### 2.1. Isolation, Enumeration, and Identification of Lactobacilli

In this study, we estimated the viability of lactobacilli in five probiotic preparations and three dietary supplements purchased from a local pharmacy. We successfully recovered viable bacteria on MRS media from six brands and did not obtain colonies with plated suspensions of the products Gm and Ne (Appendix A). Lactobacilli were isolated from the two latter brands only by employing an enrichment culture. The recovered bacteria had varied colony morphologies, two of which were particularly distinctive. Colonies of the isolates from brands At and Al were less than 1 mm in diameter, circular, flat, with irregular borders, beige or without pigment, and translucent. The lactobacilli recovered from the other brands formed small (0.5–2.0 mm in diameter) colonies that were circular to irregular, opaque, cream-white or pale, and convex, with a smooth surface and a complete edge.

Bacterial enumeration revealed that samples from brands At, Ea, Ln, and Ne had fewer bacteria than claimed by their manufacturers (Table 1). Most significantly, the sample branded as Ne did not contain viable bacteria according to our results. The amount of lactobacilli stated on the datasheet of brand Gm was only 100, limiting estimation with the method used in this study. The samples of brands Al and Ls had bacteria counts 100-fold more than those claimed by the manufacturer, and the bacteria concentration in the sample of brand Ro was 10-fold higher than the stated amount.

Although MRS agar is selective for lactobacilli, some growth of other lactic acid bacteria, such as *Leuconostoc* and *Pediococcus*, may occur [19]. Therefore, we identified bacteria recovered from all the products by MALDI Biotyper analysis (Table 1). The results confirmed correct assignment of all tested isolates to the *Lactobacillus* genus. Although brand samples Ea and Ro were annotated as multispecies, the recovered lactobacilli belonged to a single species. According to our results, 11 tested strains were assigned to *L. plantarum* species, 4 strains were assigned to *L. helveticus,* 3 strains were assigned to *L. paracasei*, and 1 strain was assigned to *L. fermentum.* We isolated *L. plantarum* species from brand samples Ln and Ro, in agreement with their datasheets, whereas samples from all the other brands contained species that differed from those claimed by their manufacturers. We did not detect *L. acidophilus*, although it was listed on the datasheets of six of the eight tested brands. These discrepancies in the composition of the products may be the result of recent progress in bacterial identification technology which has enabled more precise and accurate species identification [20].

### 2.2. Phenotypic Resistance of Lactobacilli

All *Lactobacillus* isolates from tested probiotic products were analyzed for antibiotic susceptibility by disc diffusion method and were classified as either resistant (R), moderately susceptible (MS), or susceptible (S) based on zones of growth inhibition. Between three and eight isolates recovered from each product were tested, and additional isolates of the same species with a common source and identical antibiotic resistance profiles were considered clones and were joined into one strain. Isolates of the same species recovered from a single sample were designated as separate strains when they demonstrated distinct antibiotic resistance profiles.

According to our results, most *Lactobacillus* strains were susceptible or moderately susceptible to penicillin-type antibiotics (ampicillin and amoxicillin), carbapenems (imipenem, meropenem, and ertapenem), and inhibitors of protein synthesis (chloramphenicol, erythromycin, tetracycline, clarithromycin, and linezolid) (Table 2). Based on MIC values for tetracycline, several (two *L. paracasei* strains from brand product Ea and four *L. helveticus* strains from brand product Al) susceptible to tetracycline strains were reconsidered as resistant (Appendix A). 

All tested lactobacilli were found to be tolerant of amikacin, ciprofloxacin, and norfloxacin. We also found that 79% of tested strains were resistant to vancomycin. Resistance to cephalosporins varied significantly among tested strains. Among the six cephalosporins used in this study, resistance was most frequent to ceftazidime and cefepime (95% of tested *Lactobacillus* strains). Cefoperazone and cefazolin inhibited the growth of the majority of tested strains; only 27% and 32% of strains were resistant to cefoperazone and cefazolin, respectively (Table 2).

In order to determine the role of β-lactamase in lactobacillar resistance to β-lactam antibiotics, we applied sulperazon, a well-known combination of cefoperazone and β-lactamase inhibitor sulbactam and compared the obtained inhibition halos with those induced by cefoperazone alone. We detected a significant increase in the halo size for At-2 and Ea-3, suggesting their ability to produce β-lactamases for inactivation of β-lactam antibiotics.

### 2.3. Genotypic Resistance of Lactobacilli

We used PCR amplification and subsequent DNA sequencing of the amplicons to study the presence of antibiotic resistance genes in the *Lactobacillus* strains isolated from probiotic products. The results are presented in Table 2. PCR analysis showed that only one *Lactobacillus* strain, namely *L. paracasei* Ea-1, possessed tetracycline resistance gene *tet*K. Neither another tetracycline resistance determinant (*tet*M), nor erythromycin (*erm*B*)* or chloramphenicol (*cat*) resistance genes were detected in any strain. Thirteen *Lactobacillus* strains were positive for the vancomycin resistance gene *van*X (Table 2), whereas other genes of this clusters *van*A and *van*E were not detected in any strain. We detected the gene *par*C, which is associated with resistance to ciprofloxacin, in 12 tested lactobacilli but did not obtain any PCR product for another ciprofloxacin resistance gene, *gyr*A. We also found the gene *bla*TEM, encoding extended-spectrum β-lactamase, in 16 out of 19 tested *Lactobacillus* strains. Detection of the *bla*TEM gene was confirmed by the results of the NCBI BLAST algorithm. Sequences of the resulting amplicons shared 99% similarity with the *bla*TEM gene of *Acinetobacter baumannii* (GenBank accession no. MK764360.1) and *Escherichia* sp. (GenBank accession no. NG050218.1). Other genes of β-lactams resistance, namely *bla*SHV, blaOXA-1, *bla*VIM, and *bla*IMP1, were not detected in any strain.

## 3. Discussion

Antibiotic resistance of lactobacilli, as “a double-edged sword”, can be considered ambivalently. On the one hand, intrinsic resistance is considered a favorable property of probiotic bacteria because it enables them to survive antibiotic therapy and thus prevent or treat antibiotic-associated diarrhea. On the other hand, the adverse outcome of acquired antibiotic resistance in probiotic bacteria involves the risk of its spread in the gastrointestinal microbiota and in the environment. In this context, lactobacilli used in the food industry and as probiotics can carry intrinsic resistance to a number of antibiotics but lack acquired and therefore potentially mobile antibiotic resistance genes. In this study, we aimed to investigate the antibiotic resistance of lactobacilli; therefore, we tested eight brands of *Lactobacillus*-containing probiotic preparations and dietary supplements. The viability of constituent microorganisms is often considered an essential requirement for probiotics and a prerequisite for their health benefits [5]. Estimates of the viability of lactobacilli in the investigated products revealed that in five of eight tested brands, the amount of viable lactobacilli was less than that claimed by the manufacturer. The concentration of probiotics required to obtain a clinical effect varies from 10^6^ to 10^9^ CFU per day depending on the estimates [21,22]. With the exception of Gm and Ne, samples from all brands had bacterial contents exceeding the recommended minimum threshold of 10^6^ CFU per presentation. Moreover, the enumerated lactobacilli abundance in products Al, Ln, Ls, and Ro fell within the maximum threshold of 10^9^ CFU per presentation. Although some studies suggest the efficiency of non-viable probiotics [23], it is doubtful that the claimed health benefits can be achieved by products Gm and Ne, in which we were undable to detect sufficient amounts of viable probiotic bacteria. The isolates from all the products were identified at the species level by MALDI-TOF MS, the effectiveness and reliability of which in typing lactobacilli have been confirmed in several studies [20,24]. With respect to the species content of the tested probiotic products, we detected misidentification and mislabeling in all tested brands, with the exception of Ln and Ro, which contained *L. plantarum* species contents consistent with the manufacturers’ claims. The reported overestimation of probiotic amounts in five tested products, as well as discrepancies in the species composition of six products, is consistent with previous reports concerning overestimation, misidentification, and mislabeling of food and health products containing probiotic microorganisms [25,26]. Our findings cast doubt on the accuracy and reliability of the information in the datasheets of some probiotics, substantiating the need to strengthen regulations and legislation with respect to probiotic products, which are currently largely lacking.

A total of 19 *Lactobacillus* isolates belonging to four species were obtained and investigated for their phenotypic antibiotic resistance profiles. The antimicrobials used in this work are among the top 20 of the most commonly administered antibiotics in human clinical medicine according to the data collected in 2011–2014 in SBIH “Penza regional clinical Hospital named N.N. Burdenko” (unpublished data, see Acknowledgments). The list includes chloramphenicol, erythromycin, and tetracycline, as resistances to these antibiotics are often acquired, with the potential to be transferred from lactobacilli to new hosts. 

Lactobacilli are usually susceptible to antibiotics inhibiting nucleic acid synthesis (except for fluoroquinolones) and protein synthesis (except for aminoglycosides). In this respect, resistances to fluoroquinolones and aminoglycosides are generally intrinsic to lactobacilli [11,12,13,14,15]. In our study, most *Lactobacillus* strains were found to be sensitive to chloramphenicol, erythromycin, tetracycline, clarithromycin, and linezolid and resistant to amikacin, ciprofloxacin, and norfloxacin (Table 2). These phenotypes partially coincided with the genomic program of the tested lactobacilli. The incidence of gene *par*C, which encodes the mutant form of topoisomerase IV and confers resistance to quinolones, was high among the tested *Lactobacillus* strains; however, the gene was not detected in all strains resistant to ciprofloxacin (Table 2). We also demonstrated that none of the *Lactobacillus* strains carried typical mutations in the quinolone resistance-determining region (QRDR) of the *gyr*A (DNA gyrase) gene for ciprofloxacin resistance. Hence, other mechanisms, such as efflux or cell surface impermeability, are implicated in resistance to ciprofloxacin in the strains that lacked both *par*C and *gyr*A genes [27]. 

According to disc diffusion, all tested lactobacilli showed a susceptible phenotype to chloramphenicol, erythromycin, and tetracycline (Table 2). However, based on MIC values, six strains were found to be resistant to tetracycline (Appendix A). When investigated for the presence of resistance genes, neither the chloramphenicol resistance gene *cat* of acetyl transferase, nor the most frequently observed erythromycin resistance gene, *erm*B, of 23S ribosomal rRNA methyltransferase could be amplified. Furthermore, neither of most common determinants for resistance to tetracycline in lactobacilli, gene *tet*M encoding ribosomal protection proteins and gene *tet*K encoding the tetracycline efflux pumps, were detected in any of the strains, except for *L. paracasei* Ea-1, which carried the *tet*K gene. The presence of this gene did not coincide with the phenotype of the strain, as its susceptibility to tetracycline was demonstrated by both disc diffusion and broth microdilution methods. Similarly, in our previous investigations, *Lactobacillus* strains sensitive to tetracycline were found to carry silent genes *tet*K, *tet*M, and *tet*L [28,29]. These discrepancies between the resistance phenotype and genotype may be the result of defective expression of resistance genes, as previously described in [13].

With respect to antibiotics that target the bacterial cell wall, it is well-known that most *Lactobacillus* species are sensitive to penicillin-type antibiotics and carbapenems but resistant to vancomycin. Resistance to cephalosporins is highly variable [12,13,15,30]. In the present study, 79% of the isolates were resistant to vancomycin. In an attempt to relate the observed resistance to the presence of resistance genes, we used PCR amplification of known genes encoding vancomycin resistance. Gene *van*X was detected in more than 68% of the strains, and no PCR products were amplified with *van*A and *van*E primer sets. Interestingly, three strains sensitive to vancomycin were characterized by the presence of *van*X. A wide occurrence of *van*X within lactobacilli was previously described in [31,32], as well as in our previous study [29]. Vancomycin acts by binding to the D-alanyl-D-alanine (D-ala-D-ala) moiety of the peptidoglycan precursor, blocking cell wall biosynthesis. Activity of VanX (protein D,D-dipeptidase) results in the synthesis of abnormal peptidoglycan precursors terminating in D-ala-D-lactate instead of D-ala-D-ala, eliminating the drug target. This is perhaps the best-characterized vancomycin resistance mechanism; it is considered intrinsic, generally chromosomally encoded, and not inducible or transferable in lactobacilli [11,13].

The *Lactobacillus* strains were generally sensitive to carbapenems, except for the isolates from brand product Ea, which demonstrated resistance to meropenem (strains Ea-2 and Ea-3) and ertapenem (strain Ea-1) according to disc diffusion results. Carbapenems, as a class of β-lactams, bind to penicillin-binding proteins (PBPs) and thus inhibit the synthesis of the bacterial cell wall. In vitro studies on lactobacillar susceptibility to carbapenems are scarce. In recent years, several carbapenem-resistant *Lactobacillus* isolates have been reported, attracting attention as clinical pathogens [33,34]. Even fewer data are available on the resistance of lactobacilli to cephalosporins. Although it has been established that most lactobacilli can tolerate high concentrations of cephalosporins, resistance to this class of β-lactams varies widely depending on the species and antibiotics [15,30]. In this study, cefoperazone and cefazolin demonstrated the highest activity and inhibited growth of 73% and 68% of tested strains, respectively. In contrast, ceftazidime and cefepime demonstrated the lowest activity and were ineffective against 95% of tested *Lactobacillus* strains. Understanding of the mechanisms that underlie lactobacillar resistance to carbapenems and cephalosporins remains limited. β-Lactamases represent the main mechanism of bacterial resistance to β-lactam antibiotics. Some extended-spectrum beta-lactamases (ESBLs) have been detected in clinical isolates of lactobacilli*:* OXA-48 [33], *bla*TEM, *bla*OXA-1*, bla*SHV [35], and *bla*CTX-M [36]. In our previous study, we revealed a wide distribution of the *bla*TEM gene in *Lactobacillus* isolates from different sources and detected genes *bla*SHV and *bla*OXA-1 in some of them [29]. Herein, we demonstrated the frequent occurrence of *bla*TEM in lactobacilli isolated from probiotics. TEM-type β-lactamases are usually susceptible to β-lactamase inhibitors, such as sulbactam [37]. However, many derivatives and mutants of TEM β-lactamases with inhibitor resistance have been described [38]. Among all *bla*TEM^+^ strains, only At-2 demonstrated increased halos around discs with cefoperazone/sulbactam compared to discs with cefoperazone alone, indicating the possible presence of inhibitor-susceptible β-lactamase. 

All tested strains exhibited phenotypic resistance to a number of antibiotics (4–11 antibiotic), revealing a multiple resistance pattern (Table 2). Two isolates from brand product Al were the most susceptible, exhibiting resistance to only 22% of antibiotics. In contrast, *L. plantarum* Ne isolated from brand Ne and *L. plantarum* Ro-2 from brand Ro demonstrated resistance to the most antibiotics (61% of antibiotics used in this study). Therefore, these strains were considered to be the most resistant. In addition, *L. fermentum* Ne exhibited atypical resistance to both amoxicillin and clarithromycin. These two antibiotics, along with the proton pump inhibitor, comprise a standard first-line triple therapy for *Helicobacter pylori* infection [39]. It was previously demonstrated that pretreatment with *Lactobacillus*-based probiotics may improve the efficacy of *H. pylori* eradication therapy [40]. According to our results, lactobacilli are highly susceptible to amoxicillin and clarithromycin, and only dietary supplement Na can be rationally combined with anti-*H. pylori* therapy. Overall, our data on phenotypic antibiotic resistance of probiotic lactobacilli presented in Table 2 are useful for the development of treatment schemes in which certain antibiotics are rationally combined with probiotics. Analysis of resistance genes showed that *bla*TEM (84% of strains), *van*X (68%), and *par*C (63%) were the most frequently identified in the tested *Lactobacillus* strains. We also detected potentially transferable gene *tet*K in *L. paracasei* strain Ea-1, although this strain was not found to be resistant to tetracycline.

## 4. Materials and Methods

### 4.1. Probiotics and Enumeration of Lactobacilli Contents

Eight brands of probiotic preparations and dietary supplements designated here as Al, At, Ea, Gm, Ln, Ls, Ne, and Ro were purchased from a local pharmacy; their details are listed in Table 1.

The concentration of the lactobacilli in probiotics were enumerated using the drop plate method [41]. To that end, one randomly chosen form of presentation (capsule, sachet, vial, or suppository) was dissolved in sterile physiological saline, subsequently diluted 10-fold, and dropped in 5 μL drops onto de Man, Rogosa, and Sharpe (MRS) agar (HiMedia, Mumbai, India), which is selective for *Lactobacillus*; therefore, other probiotic strains were excluded from the bacterial count. One presentation contained equal amounts of each probiotic strain. In products for which such information was not stated, equal contributions from each probiotic strain were assumed. The plates were incubated under anaerobic conditions (Anaerogas gaspack, NIKI MLT, St. Petersburg, Russia) at 37 °C for 48 h, after which the number of colonies was counted from the dilution, which contained 3–30 colonies per 5 μL drop. Data expressed in colony-forming units (CFU) per presentation are presented as the mean of three independent experiments and were compared to the values claimed by the manufacturers (Table 1). Within a presentation, the experimental data scatter did not exceed 5%.

### 4.2. Isolation and Identification of Bacteria

For isolation of lactobacilli from probiotic preparations and dietary supplements, the dissolved presentations were cultured overnight in MRS broth for enrichment of lactobacilli, after which 10-fold dilution series were prepared in physiological saline, plated onto MRS agar plates, and incubated under microaerophilic conditions in a candle jar at 37 °C. Randomly selected colonies we restreaked onto MRS agar to obtain pure cultures. 

The isolates were identified using a MALDI Biotyper system (Bruker Daltonics, Bremen, Germany), as previously described [28]. The mass spectra acquired following the manufacturer’s recommendations were compared with the reference spectra in the integrated database (version 3.2.1.1), and the resulting similarity values were expressed as log scores. According to the standard Bruker interpretative criteria, scores ≥2.0 were accepted for reliable species identification, scores ≥1.7 for reliable genus identification, and scores <1.70 were considered unreliable for identification [20].

### 4.3. Antibiotic Susceptibility Testing

Antibiotic susceptibility was assessed by the disk diffusion method, as previously described [42]. In brief, 0.5 McFarland turbidity standard inocula prepared from the overnight cultures were used to inoculate MRS agar plates. Antibiotic discs (Scientific Research Centre of Pharmacotherapy, St. Petersburg, Russia) (Appendix A) were placed on the agar surface. The plates were incubated for 48 h at 37 °C under anaerobic conditions (Anaerogas GasPak, NIKI MLT, St. Petersburg, Russia); the results were interpreted as susceptible (S), moderately susceptible (MS), or resistant (R) according to the inhibition halos around the antibiotic disks using the breakpoints presented in Appendix A. 

The minimum inhibitory concentrations (MICs) of tetracycline were determined by the broth microdilution method, as previously described [29]. Briefly, tetracycline (Sigma-Aldrich, St. Louis, MO, USA) was tested in concentration range of 0.125–256 μg/mL prepared in 96-well non-treated cell culture plates (Eppendorf) by a series of twofold dilutions in MRS broth. The inocula derived from the overnight cultures and adjusted to a turbidity equivalent to 0.5 McFarland standard were used to inoculate each well.

After 24 h incubation at 37 °C, the MICs were read as the lowest concentration of tetracycline at which visible growth was inhibited. The results were interpreted as susceptible (S), moderately susceptible (MS), or resistant (R) according to the breakpoints proposed in [43].

### 4.4. Detection of Antibiotic Resistance Genes

Total DNA was extracted from lactobacilli cells as described in [28]. The PCR mixture (25 µL) contained 50 ng of DNA, 10 pmol of each primer (Appendix A), 200 µM of dNTPs (dATP, dCTP, dGTP, and dTTP), 1 × PCR buffer (20 mM Tris-HCl, pH 8.8, 10 mM KCl, 10 mM (NH_4_)_2_SO_4_, 2 mM MgSO_4_, and 0.1% Triton X-100), and 1 unit of Taq DNA polymerase. PCR was performed in a C1000 thermal cycler (Bio-Rad Laboratories, Hercules, CA, USA) as follows: 94 °C for 5 min; 35 cycles of 94 °C for 30 s, annealing temperature (Appendix A) for 30 s, 72 °C for 30–70 s (calculated as one min for every 1000 nucleotides of the individual amplicons, Appendix A), and a final extension step at 72 °C for 7 min. The obtained PCR products were separated by electrophoresis (100 V) on 1% agarose gels and visualized by Midori Green (Nippon Genetics Europe, Düren, Germany) staining using a Gel Doc XR molecular imaging system (Bio-Rad Laboratories, Hercules, CA, USA). The amplicons of the expected size were purified with a GeneJET gel extraction kit (Thermo Fisher Scientific, Vilnius, Lithuania) according to the instructions of the manufacturer and sequenced by Evrogen JSC (Moscow, Russia). The obtained nucleotide sequences were analyzed using the Basic Local Alignment Search Tool (BLAST) algorithm and the GenBank database (National Center for Biotechnology Information).

## 5. Conclusions

Although probiotics are often claimed to promote health and are generally regarded as safe, our results call such statements into question. In two out of eight tested probiotic products, the amount of viable bacteria was below the recommended minimum threshold of 10^6^ CFU per presentation. Furthermore, we detected discrepancies between the information presented in the datasheets and MALDI Biotyper identification results of the isolates with respect to the species content of six products. It is therefore doubtful that the claimed beneficial health effects can be achieved by such products with altered composition. The results of this study raise concerns about the safety of the investigated probiotic products in terms of antibiotic resistance spread in the environment, as we detected unusual carbapenem and tetracycline phenotypic resistances in several strains, the *tet*K gene for tetracycline resistance in one isolate, and a wide occurrence of extended-spectrum β-lactamase gene *bla*TEM. Overall, our data provide evidence for extensive revision of the regulation of microorganisms for human consumption as probiotic preparations and dietary supplements. In particular, screening of antibiotic resistance and genotype-based assessment of genetic stability, as well as precise identification of bacteria at the strain level relative to that claimed by manufacturers and proper labeling information should be included in quality control standards for probiotics.

## Figures and Tables

**Table 1 antibiotics-11-01557-t001:** Information on probiotic bacteria in probiotic preparations ^a^ and dietary supplements ^b^.

Product	Country of Manufacture	Probiotic Content	Manufacturer Claim of Probiotic Amount	Enumerated Amount of Lactobacilli	Isolate (Strain) ^c^	Log Score ^d^
			(CFU/Pharmaceutical Form)			
Al ^a^	Russia	*L. acidophilus*	10^7^ per capsule	2.8 × 10^9^	*L. helveticus* Al-1 (Unchanged) *L. helveticus* Al-2 (Unchanged) *L. helveticus* Al-3 (Unchanged) *L. helveticus* Al-4 (Unchanged)	2.326 2.368 2.333 2.302
At ^a^	Russia	*L. acidophilus*	10^7^ per vaginal suppository	3.9 × 10^6^	*L. plantarum* At-1 (*Lactiplantibacillus plantarum* At-1) *L. plantarum* At-2 (*Lactiplantibacillus plantarum* At-2) *L. plantarum* At-3 (*Lactiplantibacillus plantarum* At-3)	2.342 2.399 2.436
Ea ^b^	Russia	*L. acidophilus,* *L. helveticus,* *Lactococcus lactis,* *Streptococcus thermophilus,* *Propionibacterium freudenreichii* ssp. *shermanii*	4.0 × 10^9^ per vial	1.6 × 10^6^	*L. paracasei* Ea-1 (*Lacticaseibacillus paracasei* Ea-1) *L. paracasei* Ea-2 (*Lacticaseibacillus paracasei* Ea-2) *L. paracasei* Ea-3 (*Lacticaseibacillus paracasei* Ea-3)	2.427 2.444 2.429
Gm ^a^	Bulgaria	*L. delbrueckii* ssp. *bulgaricus* 51	10^2^ per morsulus	0	*L. fermentum* Gm (*Limosilactobacillus fermentum* Gm)	2.115
Ln ^a^	Russia	*L. plantarum* 8P-A3 or *L. fermentum* 90T-C4	10^10^ per vial	3.7 × 10^9^	*L. plantarum* 8PA3 (*Lactiplantibacillus plantarum* 8PA3)	2.452
Ls ^a^	Slovenia	*L. acidophilus,* *Bifidobacterium infantis,* *Enterococcus faecium*	1.2 × 10^7^ per capsule	1.1 × 10^9^	*L. plantarum* Ls (*Lactiplantibacillus plantarum* Ls)	2.312
Ne ^b^	Armenia	*L. acidophilus*	1.8 × 10^8^ per capsule	0	*L. plantarum* Ne (*Lactiplantibacillus plantarum* Ne)	2.282
Ro ^b^	Netherlands	*L. acidophilus* (2 strains), *L. plantarum,* *L. paracasei,* *L. rhamnosus,* *L. salivarius,* *Bifidobacterium lactis*, *B. bifidum*	5.0 × 10^8^ per capsule	2.4 × 10^9^	*L. plantarum* Ro-1 (*Lactiplantibacillus plantarum* Ro-1) *L. plantarum* Ro-2 (*Lactiplantibacillus plantarum* Ro-2) *L. plantarum* Ro-5 (*Lactiplantibacillus plantarum* Ro-5) *L. plantarum* Ro-7 (*Lactiplantibacillus plantarum* Ro-7) *L. plantarum* Ro-8 (*Lactiplantibacillus plantarum* Ro-8)	2.282 2.278 2.173 2.279 2.298

^c^ New names for *Lactobacillus* species according to reclassification by Zheng J. et al. (2020) [1] are in parentheses. ^d^ MALDI Biotyper log score ≥ 2.0 indicates reliable identification at the species level.

**Table 2 antibiotics-11-01557-t002:** Antibiotic resistance of *Lactobacillus* strains and detection of antibiotic resistance genes by polymerase chain reaction (PCR).

		Phenotype ^a^																					
No.	Strain	Amikacin	Ampicillin	Amoxicillin	Imipenem	Meropenem	Ertapenem	Cefazolin	Cefotaxime	Ceftazidime	Ceftriaxone	Cefoperazone	Cefoperazone/sulbactam	Cefepim	Clarithromycin	Vancomycin	Linezolid	Ciprofloxacin	Norfloxacin	Erythromycin	Chloramphenicol	Tetracycline	Genotype
1	*L. plantarum* 8PA3	R	S	S	S	S	S	S	R	R	MS	S	S	R	S	S	S	R	R	MS	S	S	-
2	*L. fermentum* Gm	R	S	S	S	S	S	S	MS	R	MS	MS	S	R	S	S	S	R	R	MS	S	S	*van*X, *bla*TEM
3	*L. plantarum* Ne	R	S	R	S	S	S	MS	R	R	R	MS	R	R	R	R	S	R	R	S	S	S	*van*X, *bla*TEM
4	*L. plantarum* Ro-1	R	S	S	S	S	S	MS	R	R	R	R	R	R	S	R	S	R	R	S	S	MS	*par*C, *bla*TEM
5	*L. plantarum* Ro-2	R	S	S	S	S	S	R	R	R	R	R	R	R	S	R	S	R	R	S	S	S	*par*C, *bla*TEM
6	*L. plantarum* Ro-5	R	S	S	S	S	S	S	R	R	R	R	MS	R	S	R	S	R	R	S	S	MS	*van*X, *par*C, *bla*TEM
7	*L. plantarum* Ro-7	R	S	S	S	S	S	MS	R	R	R	MS	R	R	S	R	S	R	R	S	S	MS	*parC*
8	*L. plantarum* Ro-8	R	S	S	S	S	S	S	R	R	MS	R	R	R	S	R	S	R	R	S	S	MS	*van*X, *par*C, *bla*TEM
9	*L. plantarum* At-1	R	S	S	S	S	S	MS	R	R	R	R	R	R	S	R	S	R	R	S	S	MS	*van*X, *par*C, *bla*TEM
10	*L. plantarum* At-2	R	S	S	S	S	S	S	R	R	R	MS	S	R	S	R	S	R	R	S	S	S	*van*X, *par*C, *bla*TEM
11	*L. plantarum* At-3	R	S	S	S	S	S	MS	MS	R	R	MS	R	R	S	R	S	R	R	S	S	MS	*par*C, *bla*TEM
12	*L. plantarum* Ls	R	S	S	S	S	S	S	R	R	MS	MS	MS	R	S	R	S	R	R	S	S	MS	*van*X, *bla*TEM
13	*L. paracasei* Ea-1	R	MS	S	MS	MS	R	R	MS	R	R	MS	MS	R	S	R	S	R	R	S	S	S	*van*X, *bla*TEM, *tet*K
14	*L. paracasei* Ea-2	R	MS	S	S	R	MS	R	MS	R	R	MS	MS	R	S	R	S	R	R	S	S	S	*bla*TEM
15	*L. paracasei* Ea-3	R	R	S	S	R	MS	R	MS	R	R	MS	S	R	S	R	S	R	MS	S	S	S	*van*X
16	*L. helveticus* Al-1	R	S	S	S	S	MS	R	S	R	MS	MS	MS	R	S	R	S	R	R	S	S	S	*van*X, *par*C, *bla*TEM
17	*L. helveticus* Al-2	R	S	S	S	S	MS	R	MS	R	MS	MS	MS	R	S	R	S	R	R	S	S	S	*van*X, *par*C, *bla*TEM
18	*L. helveticus* Al-3	R	S	S	S	S	S	S	S	R	S	S	S	MS	S	S	MS	R	R	S	S	S	*van*X, *par*C, *bla*TEM
19	*L. helveticus* Al-4	R	S	S	S	S	S	S	S	S	S	S	S	R	S	S	S	R	R	S	S	S	*van*X, *par*C, *bla*TEM

^a^ Based on standards mentioned in Materials and Methods, lactobacilli were characterized as either susceptible (S), moderately susceptible (MS), or resistant (R) to each antibiotic.

## Data Availability

The data used to support the findings of this study are included within the article.

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
