# Peer review of "Alarming Antibiotic Resistance of Lactobacilli Isolated from Probiotic Preparations and Dietary Supplements"

_antibiotics, 2022, doi:10.3390/antibiotics11111557_

Round 1
Reviewer 1 Report
In the present study entitled "Alarming antibiotic resistance of Lactobacillus spp. isolated from probiotic preparations and dietary supplements," the authors approach a topic of wide interest - probiotics in different formulations being widely used for various reasons.
The "Introduction" is well documented and organized. As a substantial warning signal, the authors could mention the latest case reports regarding Lactobacilli infections (Rossi et al., 2022).
The "Results" are presented in a detailed manner. The Lactobacilli isolated from several formulations currently used as probiotics elevate the readers' interest. The presentation of phenotypic and genotypic resistance to Antibiotics increases the value of this study and supports the conclusions.
"Discussion" is an extended section; all aspects are pointed out with suitable references.
"Material and Methods" are well structured.
"Conclusion" synthesized the most important aspects of this study for revealing the inadvertence between the information in the datasheets and the results of the isolates' identification. Moreover, their data evidence the need for a rigorous revision of the regulation regarding probiotic products with microorganisms for human consumption.
Author Response
Comment:
The "Introduction" is well documented and organized. As a substantial warning signal, the authors could mention the latest case reports regarding Lactobacilli infections (Rossi et al., 2022).
Response:
We deeply appreciate the time and efforts by the Reviewer in consideration of our manuscript. We agree with this comment, and added the information and reference that the Reviewer suggested (page 2, lines 52-53 and reference 8).
Reviewer 2 Report
The article "Alarming antibiotic resistance of Lactobacillus spp. isolated from probiotic preparations and dietary supplements" is good work and a well-written manuscript. The article describes the multidrug-resistant probiotic strains from commercial drinks and dietary supplements for human consumption that can raise the concern of horizontal gene transfer.
It would be great if the authors could clarify a few concerns.
Introduction:
In the introduction, the authors mentioned that Lactobacillus is reclassified into 25 genera, but the authors used the former classification for this paper. Isn't it better to mention the new classification?
Results:
The valuable information about commercial probiotics and the basis for selecting commercial probiotics has been mentioned nowhere. However, it will be considered if the authors consider giving proper validation and justification for selecting the commercial probiotics used in this study.
It is strange not to see the name of the commercials. Is it illegal to use, or will it not be better to prove the pros and cons of the commercial scientifically? Also, it allows the companies to take proper action before marketizing something new without many scientific validations.
Authors have mentioned an increase in the CFU count in some of the commercial probiotics. Is there any proper justification or reason that authors can think off?
Methods:
The methods section is almost plagiarised from the references authors have referred to; kindly work on it.
Line 348: The number of bacteria(CFU/mL) used for MIC is not according to CLSI guidelines.
Author Response
The article "Alarming antibiotic resistance of Lactobacillus spp. isolated from probiotic preparations and dietary supplements" is good work and a well-written manuscript. The article describes the multidrug-resistant probiotic strains from commercial drinks and dietary supplements for human consumption that can raise the concern of horizontal gene transfer.
It would be great if the authors could clarify a few concerns.
Response: We deeply appreciate the time and efforts by the Reviewer in consideration of our manuscript. In accordance with the given comments, the manuscript has been improved. In addition to submitting an altered version of the manuscript, we are providing a point-by-point response to the Reviewer’s comments outlined below.
Comment 1: Introduction:
In the introduction, the authors mentioned that Lactobacillus is reclassified into 25 genera, but the authors used the former classification for this paper. Isn't it better to mention the new classification?
Response 1: We are grateful to the Reviewer for bringing this important point to our attention. There are several reasons why we do not use the new nomenclature in this paper:
1) We compare the species isolated from probiotic preparations and dietary supplements with those claimed by the manufacturers. Manufacturers use the old nomenclature in the datasheets.
2) For species identification we used MALDI-TOF mass spectrometry (Bruker Biotyper system, Bruker Daltonics, Germany); its integrated database (version 3.2.1.1) relies on the old nomenclature.
Yet, new nomenclature was presented in Table 1 (footnote “c”). Following your recommendation, we replaced Lactobacillus (genus Lactobacillus, Lactobacillus spp.) by “lactobacilli” where it was possible throughout the manuscript.
C2: Results:
The valuable information about commercial probiotics and the basis for selecting commercial probiotics has been mentioned nowhere. However, it will be considered if the authors consider giving proper validation and justification for selecting the commercial probiotics used in this study.
R2: In this study we investigated the lactobacilli components and quantified their amount and viability in eight commercially available brands of Lactobacillus-containing probiotic preparations and dietary supplements with various pharmaceutical forms (formulations), from various geographical areas and thus various producers.
C3: It is strange not to see the name of the commercials. Is it illegal to use, or will it not be better to prove the pros and cons of the commercial scientifically? Also, it allows the companies to take proper action before marketizing something new without many scientific validations.
R3: Thank you. The comment is reasonable and we appreciate the opportunity to clarify our view. Our work was inspired by (Wong A. et al., 2015) where authors designate five tested commercially available dietary supplements as follows: Bi, Bn, Bg, Cn and L. We suppose that it’s illegal to give full names of the commercial probiotics in the scientific paper because production of these goods and testing of their quality are regulated by special documents, like national standards, methodological (procedural) guidelines, and regulations. Although they are based on the general principles of lactobacilli cultivation and enumeration, still are not identical to the methods used in this paper. Besides, they are often different in different countries or groups of countries (e.g. customs union). In other words, the quality of every probiotic should be tested using the national standards, methodological (procedural) guidelines, and regulations of the country in which it was manufactured. Hence, according to the current regulation results of our paper do not have legal validity to estimate the quality of commercial probiotics.
Wong A, Ngu DY, Dan LA, Ooi A, Lim RL. Detection of antibiotic resistance in probiotics of dietary supplements. Nutr J. 2015 Sep 14;14:95. doi: 10.1186/s12937-015-0084-2. PMID: 26370532; PMCID: PMC4568587.
C4: Authors have mentioned an increase in the CFU count in some of the commercial probiotics. Is there any proper justification or reason that authors can think off?
R4: The lactobacilli concentrations in the samples of the brands Al and Ls were 100-fold more than that claimed by the manufacturers, while the lactobacilli count in the sample of brand Ro was 10-fold higher than the stated amount. But still the enumerated lactobacilli abundance in the products Al, Ls, and Ro fell within the maximum threshold of 109 CFU per presentation. Moreover, according to the National Institutes of Health probiotic supplements may contain up to 10 billion CFU per dose (https://ods.od.nih.gov/factsheets/Probiotics-HealthProfessional/). Current regulations do not bar the manufacturers from elevating of probiotic bacteria count in preparations, so that the number of CFU at the end of the product’s shelf life fall within the desirable range from 106 to 109 CFU. However, further increase in the content of bacteria in the preparations entails unnecessary expenses (is unprofitable) for manufacturers.
C5: Methods:
The methods section is almost plagiarised from the references authors have referred to; kindly work on it.
R5: Thank you. The comment is reasonable and we appreciate the opportunity to improve the manuscript in accordance with your recommendations.
C6: Line 348: The number of bacteria(CFU/mL) used for MIC is not according to CLSI guidelines.
R6: Thank you, it has been corrected.
